# Role of Combined Na_2_HPO_4_ and ZnCl_2_ in the Unprecedented Catalysis of the Sequential Pretreatment of Sustainable Agricultural and Agro-Industrial Wastes in Boosting Bioethanol Production

**DOI:** 10.3390/ijms23031777

**Published:** 2022-02-04

**Authors:** Shaimaa Elyamny, Ali Hamdy, Rehab Ali, Hesham Hamad

**Affiliations:** 1Electronic Materials Research Department, Advanced Technology and New Materials Research Institute (ATNMRI), City of Scientific Research and Technological Applications (SRTA-City), Alexandria 21934, Egypt; selyamny@srtacity.sci.eg; 2Environmental Biotechnology Department, Genetic Engineering and Biotechnology Research Institute (GEBRI), City of Scientific Research and Technological Applications (SRTA-City), Alexandria 21934, Egypt; ali.hamdy343@yahoo.com; 3Fabrication Technology Research Department, Advanced Technology and New Materials Research Institute (ATNMRI), City of Scientific Research and Technological Applications (SRTA-City), Alexandria 21934, Egypt

**Keywords:** waste recycling, Tetra Pak, corn stover, pretreatment, ZnCl_2_, phosphate, functionalization, enzymatic hydrolysis, fermentation, bioethanol production

## Abstract

Improper lignocellulosic waste disposal causes severe environmental pollution and health damage. Corn Stover (CS), agricultural, and aseptic packaging, Tetra Pak (TP) cartons, agro-industrial, are two examples of sustainable wastes that are rich in carbohydrate materials and may be used to produce valuable by-products. In addition, attempts were made to enhance cellulose fractionation and improve enzymatic saccharification. In this regard, these two wastes were efficiently employed as substrates for bioethanol production. This research demonstrates the effect of disodium hydrogen phosphate (Na_2_HPO_4_) and zinc chloride (ZnCl_2_) (NZ) as a new catalyst on the development of the sequential pretreatment strategy in the noticeable enzymatic hydrolysis. Physico-chemical changes of the native and the pretreated sustainable wastes were evaluated by compositional analysis, scanning electron microscopy (SEM), X-ray diffractometry (XRD), Fourier transform infrared spectroscopy (FTIR), and thermogravimetric analysis (TGA). These investigations showed major structural changes after the optimized sequential pretreatment. This pretreatment not only influences the delignification process, but also affects the functionalization of cellulose chemical structure. NZ released a higher glucose concentration (328.8 and 996.8 mg/dl) than that of ZnCl_2_ (Z), which released 203.8 and 846.8 mg/dl from CS and TP, respectively. This work led to the production of about 500 mg/dl of ethanol, which is promising and a competitor to other studies. These findings contribute to increasing the versatility in the reuse of agricultural and agro-industrial wastes to promote interaction areas of pollution prevention, industrialization, and clean energy production, to attain the keys of sustainable development goals.

## 1. Introduction

Shockingly, a growing developing populace and innovative advancements have led to an increment in solid wastes. Agricultural wastes such as rice straw, wheat straw, and corn stover are attractive to be used as a lignocellulosic feedstock for biofuel production instead of traditional disposal by transportation to landfills, incineration, and composting [1].

Corn Stover (CS) is one of the lignocellulosic materials that have high productivity, local availability, and almost no cost. The global annual corn production is about 1 billion tons [2]. Two-thirds of the corn crop is a non-valuable which is considered as a solid waste that is so far over any local uses and leads to incorrect disposal problems. CS is always burnt in an open area and contributes to significant harmful effects on global warming, climate change, and human health as it leads to skin and eye irritation, neurological, cardiovascular, and respiratory diseases, such as lung infections, coughs, and bronchitis [3].

In another way, Tetra Pak (TP), as an example of agro-industrial wastes, multi-layer poly-coated paperboards named after their first and largest manufacturer, is widely used as aseptic packages for beverages such as milk and juice [4]. It is currently estimated that two thirds of the World’s food is packed with TP packages, and around 183 billion TP packages were distributed in more than 160 countries in 2020 [5]. As a result, TP creates part of the significant problem of municipal solid waste (MSW). TP is made up of 25% low-density polyethylene (LDPE), 70% kraft paper, and 5% aluminum, all of which are laminated in six layers. TP is made from high-quality materials and serves as a valuable resource for recovering components that can be recycled and reused. In 2018, the companies of TP had a global recycling rate of 26% [6]. TP waste recycling is required to diminish the environmental impact of packaging industry waste output, and it can also contribute to a circular economy if the valuable elements in TP can be reused or recycled, thus conserving natural resources.

As a result, utilization of biomass resources and solid wastes based on the agricultural (e.g., CS) and agro-industrial (e.g., TP) wastes for developing biomass-derived biofuels to address the shortage of fossil resources has become imperative.

Lignocellulosic biomass is used for second-generation biofuel production that has several positive impacts on three fundamental pivots: (1) the economic pivot, as it is sustainable, increases investment in plant and equipment, leads to agricultural development, increases the number of rural manufacturing jobs, and reduces the dependency on imported fuels; (2) the energy security pivot, as it is renewable, available, and reduces the use of fossil fuels; and (3), the environmental pivot, as it is biodegradable, improves land and water use, leads to high combustion efficiency, reduces greenhouse gases (GHG), and consequently reduces air pollution [7].

Lignocellulosic biomass is composed of approximately 38–50% cellulose, 23–32% hemicellulose, and 15–25% lignin [8]. Cellulose, the wealthiest division in lignocellulosic biomass, is a promising precursor that can be transformed into bioethanol by chemical and/or biological methods. Chains of D-glucose units linked by β–1,4 glycosidic bonds comprise the chemical structure of cellulose, resulting in D-glucopyranose. Because of strong intramolecular and intermolecular H-bonds and van der Waals interactions, these chains are organized, resulting in cellulose being a semicrystalline polymer with coexisting amorphous and crystalline structures [9]. This structure allows very low accessibility of reactants or enzymes, providing high chemical stability to the cellulose. On the other hand, cellulose is strictly covered by lignin and hemicelluloses which are hardly degraded to facilitate cellulose fractionation [10]. As a result, pretreatment is an essential process to functionalize and fractionate cellulose and to remove hemicelluloses and lignin which functions as inhibitors to the enzyme during the enzymatic processes.

Many strategies of lignocellulose pretreatment including carboxylic acid treatment, ammonia fiber explosion, microbial delignification, steam explosions, ionic liquid, organosolvent, alkali pretreatment, acid pretreatment, ozonolysis, subcritical water, and supercritical CO_2_ [11,12] have been examined. Biological pretreatment is one of the most efficient lignocellulosic pretreatment methods. The main benefits of biological methods include no chemical recycling after pretreatment, simple operation, minimum inhibitor formation, low downstream processing costs, and low energy consumption. However, the hydrolysis low rate is the main obstacle in developing this method [13]. Inorganic salts and organic acids have been proven to boost the degradation of hemicelluloses and lignin, with inorganic salts being especially appealing since they are less corrosive than inorganic acids. Moreover, these salts have been investigated in combination with various chemicals which include acids, organosolvents, ionic liquids and other inorganic salts [12].

Inorganic salts can be classified into alkalic [14] or metal salts [15]. Alkalic salts have been reported as viable substitute catalysts for alkali-based pretreatments like sodium hydroxide (NaOH). These include Na_3_PO_4_ × 12H_2_O, Na_2_CO_3_, Na_2_S and Na_2_CO_3_.1.5H_2_O_2_ [15,16,17]. This salt type causes lignin and hemicellulose dissolution, lignin restructuring and conversion, alteration of the crystalline state of cellulose, as well as functionalization of cellulose by various phosphor-containing groups [18,19,20]. Qing et al. [14] found that under milder pretreatment conditions, such as lower pretreatment temperature and lower alkali loading, Na_3_PO_4_ has a greater ability to extract lignin from lignocellulosic biomass than Na_2_CO_3_. Other types of metal salts such as sulfates, phosphates, and chlorides are reported in lignocellulosic pretreatment [21,22,23]. The effects of NaCl, KCl, CaCl_2_, ZnCl_2_, and FeCl_3_ on the pretreatment of Miscanthus straw were investigated and proved that while FeCl_3_ removed 100% of the xylan, ZnCl_2_ released a larger glucose yield (90%) compared to FeCl_3_ (55%) [15].

To date, scarce applications have been proposed for the pretreatment using a combination of alkalic and metal salts for pretreatment of agriculture and/or agro-industrial wastes [24]. Indeed, to be the best of the authors’ knowledge, no study on pretreatment using ZnCl_2_ (Z), and the combined Na_2_HPO_4_ –ZnCl_2_ (NZ) in presence of autoclaving for bioethanol production has been conducted. Accordingly, the feasibility of the pretreatment using these salts was evaluated for two kinds of wastes; one is an example of agricultural waste, Corn Stover, and the other is an example of agro-industrial waste, Tetra Pak. To provide more information about the physico-chemical changes that occurred through these pretreatment processes, the compositional structure, crystallinity, functional groups, and surface morphology of untreated and pretreated CS and TP have been compared. Moreover, their effect on glucose production yields in subsequent enzymatic hydrolysis has been investigated. Consequently, their effect on bioethanol production yield was investigated as well. Figure 1 shows the schematic diagram of the pretreatment of TP and CS by Z and NZ for enzymatic hydrolysis and subsequently bioethanol production.

## 2. Results 

Lignin and hemicellulose, present in the plant cell wall to cover and protect the cellulose, obstructs the cellulose fractionation, and hinders the enzymatic hydrolysis that converts the cellulose to glucose, which consequently leads to the production of bioethanol by further processing. Hence, in this work, the CS and TP were pretreated using different salts: ZnCl_2_ as a transition metal salt to produce CSZ and TPZ, respectively, and alkalic and metal salt (Na_2_HPO_4_ and ZnCl_2_) to produce CSNZ and TPNZ, respectively, with the assistance of autoclave to break the TP and CS complex structure. The effect of these salts on the physico-chemical properties, the enzymatic hydrolysis, and further bioethanol production has been investigated.

### 2.1. Correlation between Composition Variance and Enzymatic Hydrolysis of the Pretreated Samples

The chemical composition (hemicellulose, cellulose, and lignin) of TP, CS, and their pretreated samples (TPZ, TPNZ, CSZ, and CSNZ) are analyzed and listed in Table 1. The solid recovery, cellulose recovery, hemicellulose removal, and lignin removal of each substrate using the two pretreatments are detected and summarized in Figure 2. Changing the chemical pretreatment for each substrate resulted in different solid recovery and compositions, reflecting the different effects of each pretreatment process. The substrate solid recovery, which is connected mainly with the sample weight loss, has resulted from the components solubilization, such as hemicellulose, lignin, and cellulose. The high solid recovery associated with high cellulose recovery indicates one of the advantages of the pretreatment using these salts. However, the pretreatment using Na_2_HPO_4_-ZnCl_2_ led to higher solid recovery, lignin removal, and cellulose recovery than the pretreatment using ZnCl_2_ only for the two substrates. The solid recovery in the case of TPNZ and TPZ samples was higher than the solid recovery in the case of CSNZ and CSZ, which may be related to the higher lignin and hemicellulose content in the CS, which are dissolved and leads to a lower solid recovery. The recovery of cellulose fraction was higher than 90% for TP and CS substrates. This means that the pretreatment using Na_2_HPO_4_-ZnCl_2_ is safer and more protective of cellulose, which indicates high selective solubilization. The lignin removal was between 31.2 and 37% in all samples, except in the case of the TPNZ sample (67.1%), which may be related to the lignin fiber weakness due to the industrial processing of the T substrate which facilitated the lignin dissolution and revealed the effectiveness of the Na_2_HPO_4_–ZnCl_2_ pretreatment.

The efficiency of the pretreatment process was reflected in the enzymatic hydrolysis, which was conducted using CS, TP, and their pre-treated samples (Figure 3). The delignification process always reflects the cellulose fractionation and accessibility, and increased the enzymatic digestibility [14]. Hence, the TPNZ sample has the highest glucose production yield as it has the highest delignification. Furthermore, lignin removal increased the substrate surface porosity. In this study, the increase in the enzymatic hydrolysis efficiency almost exclusively resulted in improvements in the glucose content, and the enhancement of functional groups (attributable to the addition of Na_2_HPO_4_-ZnCl_2_), in addition to the nature of raw material (TP is better than CS). The higher lignin content in CSZ and CSNZ than TPZ and TPNZ may be the reason for the higher cellulose protection, lower cellulose fractionation, and lower glucose yield [25].

Generally, the improvement in the produced glucose after pretreatment may be related to the low lignin content, which implies exposure of a higher number of active sites and the 1-beta glucoside bonds of the cellulose to the enzymes [26]. These results correspond with those found in the literature [27].

As shown in Figure 3, the enzymatic hydrolysis was enhanced by 6 and 1.47 times compared with the untreated CS and TP by applying the pretreatment using ZnCl_2_, respectively. This improvement can be explained by one of the possible mechanisms of the cellulose fractionation/transformation and can subsequently enhance the enzymatic hydrolysis. Firstly, zinc salts make the crystalline cellulose of CS and TP more swollen and easier to degrade by the action of the enzyme [28]; secondly, ZnCl_2_ acts as a Lewis acid that can activate specific functional groups and cleave the C–O bond during the depolymerization process, especially the glycosidic connections within lignocellulosic structures [23]; finally, the Cl atom is a high electronegative element, hence, the Cl anion is an excellent hydrogen bond acceptor for lignin and a polarization reagent for C–O bonds, and it has been utilized in biomass dissolution transformation [28].

On the other hand, combining Na_2_HPO_4_ with ZnCl_2_ for pretreatment of CS and TP enhanced the enzymatic hydrolysis by 9.75 and 1.73 times compared with untreated CS and TP, respectively. This improvement is a result of the dual action of Na_2_HPO_4_ with ZnCl_2_. Moreover, this pretreatment exhibited higher improvement during the enzymatic hydrolysis for the two substrates due to its high buffering capacity which encourages the formation of acidic compounds including acetic acids from hemicelluloses and organic acids from lignin, which lead to the destruction of lignin and facilitates enzyme accessibility [26]. The saccharification of TP, TPZ, and TPNZ typically led to the production of 94, 96, and 96% based on the theoretical yield of 0.51 g ethanol/g glucose with ethanol productivity of 0.13 g/L/h. The previous results proved that using metal salts like ZnCl_2_ and Na_2_HPO_4_ can boost bioethanol production from the raw substrate, as shown in Figure 3. In order to compare the ethanol production by various feed stocks and pretreatment and fermentation conditions, Table 2 shows that the superior bioethanol production by ZnCl_2_ and Na_2_HPO_4_ was clearly confirmed when compared to the literature [11,29,30,31,32,33].

### 2.2. Surface Morphology

To explore the influence of the pretreatment on the surface morphology and the physical features changes of CS and TP after pretreatment with Z and NZ, SEM analysis was investigated, and is presented in Figure 4. The SEM images of the untreated TP sample show a rough surface with chunky aggregates (Figure 4a), while the untreated CS exhibited smooth, compact, wall formed with thick-walled fiber cells, without any pores or cracks, which indicates high lignin covering and cellulose protection, and biomass recalcitrance (Figure 4d). This obvious difference between the two surfaces may be the direct reason for the higher response of the TP sample to the NZ pretreatment than the CS sample and leads to high delignification, as mentioned in Section 3.1. After the pretreatment process, the morphology of TPZ and TPNZ is converted to longitudinal, continuous, separated, well-defined fibers (Figure 4b,c), respectively. The surface morphology of TPNZ is almost similar to TPZ, but the fibers are thinner. These fibers could be generated as a result of the performed pretreatments which confirm the delignification process. On the other hand, the morphology of CSZ shows that the cell structure was disrupted, and the cell interior structure was disclosed (Figure 4e), while CSNZ shows the structure of large, aggregated particles with pores and the microstructure becomes more disorganized. This morphology was characterized by the loosening of the massive fibrous network with greater exposure of fibers, which might be due to the solubilization of cell wall components as a result of the effect of Na_2_HPO_4_ (Figure 4f).

TP samples, after pretreatment, consisted of individual fibers with high homogeneity, while in the case of CS samples, they consisted of larger fiber bundles and irregular aggregates. It is worth noticing that the pretreatments of TP samples that are used in this work can separate the fibers rather than pretreated CS samples, and visually show the unique morphological changes after using the Z pretreatment and the higher homogeneity after the NZ pretreatment, which is beneficial for enzymatic hydrolysis.

After pretreatment, the SEM images show that the cell structure was damaged, which exposed the cells’ inner contents with observed fibers, especially in TP samples. It was supposed that this breaking of the crystallized cellulose–hemicellulose–lignin structure and the significant removal of lignin increased surface exposure to the enzymes, and thereby enhanced the accessibility of enzymes to cellulose, resulting in increasing enzymatic digestibility and increased bioethanol production yield [34]. By comparing the TPNZ with CSNZ, is the fibers are clearly maintained, indicating that the pretreatment of CS destroyed the structure of the skeleton of CS after pretreatment with ZnCl_2_ and Na_2_HPO_4_, hence the glucose yield produced from pretreated CS was lower than that produced from pretreated TP. In addition, the thinner fibers of TPNZ than TPZ were favorable for the enzymatic hydrolysis and leads to a higher glucose yield and consequently a higher bioethanol production yield. These observations are supported by those results found in Table 1 and Figure 3.

### 2.3. Chemical Structure

In order to investigate the functional groups attached to the TP, CS, and their pretreated samples, FTIR spectra are detected qualitatively, as shown in Figure 5a,b. Various bands were detected to monitor the chemical changes that occurred in the lignocellulosic biomass after pre-treatment, and new peaks were observed, indicating that the pretreatment introduced new chemical groups into cellulose. The intensities of the bands related to cellulose increase by increasing the cellulose fractionation. The position of bands is influenced by the change of inter- and intra-molecular hydrogen bonding, and therefore is related to changes in the chemical surface groups and crystallinity [6].

The samples showed the broad band at 3323 cm^−1^ which is corresponds to the hydrogen bond stretching bands of O–H symmetric and asymmetric stretching vibration, which possibly originate from the presence of glucoside linkages of cellulose, alcohols, phenols, hydroxyphenyl, guaiacyl, and syringyl groups of lignin and/or physisorbed and chemisorbed water molecules in cellulose [35,36]. However, these absorption bands gradually became broader, especially with pretreatment by ZnCl_2_ alone, which could be related to the weakness of hydrogen bonding between cellulose chains, while more intense bands exist in the samples treated with Na_2_HPO_4_–ZnCl_2_ as a result of the change of crystallinity by phosphor-containing groups [6,37]. The peaks at 2900 cm^−1^ are corresponding to C–H stretching vibrations and indicate the presence of hemicellulose and/or lignin [3,38,39]. It is worth noticing that this peak disappeared in the TPZ sample, which indicates significant lignin and/or hemicellulose removal. The peak at 2349 cm^−1^ that appeared in all samples is related to the –OH stretching vibration mode of the hydroxyl functional groups [40,41]. Taking this into account, these bands are more intense for the samples treated by ZnCl_2_ alone, indicating the introduction of more oxygen-functional groups in TPZ and CZ. C=O stretching vibration of either the ester linkage of the carboxylic group the ferulic and p-coumaric acids of lignin, pectin, and/or hemicellulose, or uronic and acetyl ester groups of the hemicelluloses, is represented by the peak at 1732 cm^−1^, which appeared in CS and TP samples, while it disappeared in the pretreated samples [42,43]. This may point to the efficient lignin and hemicellulose removal. C=O stretching and C=C aromatic skeletal stretching vibrations of lignin were located at 1630 cm^−1^ in TP and CS samples [44]. The bands at the region of 1600–1700 cm^−1^ are associated with the development of C–O and C=O groups in carboxylic, lactones, or ketone groups. Furthermore, the band at 1437 cm^−1^ is attributed to the –CH_2_ stretching vibration in CS and pretreated samples, whereas the –CH_2_ stretching vibration was weakened and shifted in the pretreated TP samples, indicating that the intramolecular hydrogen bond at C_6_ was destroyed [45]. After the pretreatment process, the intensity of these peaks is reduced, indicating lower lignin content. In the case of the CS sample, there is a peak at 1232 cm^−1^, which corresponds to C–O stretching vibration and guaiacyl/syringyl ring in xylan and/or lignin [46]. This peak vanished in the pretreated samples CSZ and CSNZ, which means that the pretreatment leads to effective delignification for the CS sample. The very sharp peaks of high-intensities observed at 1030 cm^−1^ are attributed to C–OH bending in β –(1,4)-glycosidic linkages between glucose and cellulose, C_1_–H deformation with ring vibration, C–C stretching, and C–O–C, which are the cellulose characteristics peaks for pyranose ring skeletal vibration [8]. In TPNZ and CSNZ, the intensity of this band is more intense than TPZ and CSZ samples, especially in TPNZ, indicating the contribution of P–OH groups stretching vibrations, P–O–C carbons asymmetric stretching, interactions between aromatic ring vibration and P–C (aromatic) stretching, and/or symmetrical stretching P=O in PO_2_ and PO_3_ phosphate–carbon complexes [6]. The bands at about 560 and 600 cm^−1^ have appeared in the TPNZ and CSNZ samples, and these bands are related to the presence of the PO_4_^−3^ group, which indicates successful functionalization [47]. The spectra of TPZ and CSZ are similar to those of raw materials, TP and CS hence, no new chemical group is produced after pretreatment using ZnCl_2_, indicating that it is a non-derivative solvent. The Z and NZ pretreatments decreased the aromatic content and increased the oxygen functional groups, which revealed the high delignification efficiency, which might significantly boost the number of active sites on the surface and improve the enzymatic hydrolysis.

### 2.4. Surface Functional Groups

Figure 6 depicts the quantitative data obtained from Boehm’s acid-base titration for acidic functionalities on TP and CS samples before and after pretreatment. Oxygen-containing functional groups such as carboxylic, phenolic, and lactonic groups are formed on the surface of TP and CS during the pretreatments. Hence, it evaluates the degradation of lignin structure by the determination of these surface functional groups [26]. These results confirmed the successful delignification as a result of the lower percentage of surface carboxylic groups and the higher percentage of phenolic and lactonic groups for the pretreated samples than untreated samples of TP or CS. The NZ modification has the highest amount of phenolic hydroxyl group content, which confirms the successful delignification process with high proton levels and enhanced the loss of terminal groups in the lignin structure [48]. Moreover, it has the lowest carboxylic content, which confirms the removal of acidic lignin and acidic hemicelluloses. At the same time, the surface of the basic group was found to decrease the possibility as compared to the acidic surface group, which indicated that the enhancement of oxygen content leads to a reduction of the electronic density of basal lines, and consequently, diminished the Lewis types of basic sites which associated with the π-electron-rich region that is obtained on basal lines [49]. These results were consistent with FTIR and elemental analysis results.

### 2.5. Crystal Structure

The changes in the microstructure of TP, CS, and Z and NZ pretreated samples have a strong influence on the physicochemical characteristics that were analyzed by XRD. As shown in Figure 7, the patterns show the emergence of two main peaks at 2θ, around 19.97° and 21.9°, which are assigned to (101) and (002) planes, respectively, and indicate the presence of a cellulose II crystalline structure without the existence of other peaks associated with the cellulose as a main component in the samples, especially in the pretreated samples [50]. Compared with the original TP and CS, the pretreated samples show different intensities of diffraction peaks, indicating the changes in the relative crystalline amounts. The crystallinity of CSZ is better than TPZ, indicating that the oxidation process is better by ZnCl_2_ with CS than TP. From our previous work, the original crystal structure of CS and TP was destroyed by phosphate groups during the NZ pretreatment, and it appeared as amorphous without any crystalline peaks as a result of reducing the number of the intermolecular hydrogen-bonding networks of cellulose during the oxidation process [6]. The peak intensities were slightly weaker in the samples that were pretreated by Na_2_HPO_4_, which could depict the successful functionalization without lowering the crystalline cellulose content, as proved in the FTIR analysis results in Section 2.2 [51]. These results indicate an enhancement and more organization in the cellulose content and effective hemicellulose and lignin dissolution. Furthermore, the reduction of TPNZ and CSNZ crystallinity improved the accessibility for functionalization by phosphate groups and an oxygen-containing group as a result of the reaction of phosphate groups which markedly destroyed the crystal regions in cellulose and affect the rearrangement of cellulose macromolecule [6]. On the other hand, the Z pretreatment leads to cellulose oxidation, which maintained or improved the TPZ and CSZ crystallinity. Interestingly, the crystallinity of TPNZ is lower than CSNZ, indicating the influence of phosphorylation than oxidation and more functionalization of phosphate groups rather than oxygen-containing groups. Moreover, TPZ and CSZ have a lower crystallinity than TP and CS, which is mainly caused by a series of reactions between Zn^2+^ and cellulose molecules that lead to rapid depolymerization of cellulose during the pretreatment [42]. In addition, the crystallinity reduction by disrupting highly ordered hydrogen bonds in the crystalline cellulose fibers has been reported by Raghavi et al. [52].

### 2.6. Thermal Stability

Due to the significant physico-chemical properties and subsequent enzymatic hydrolysis of TP samples when compared to C samples, we aimed to study the stability and organic polymer decomposition of TP, TPZ, and TPNZ, to assess the possibility for various applications; this was explored using the thermogravimetric analysis (TGA) technique, as shown in Figure 8a. Besides, DTG is subjected to measure the rate at which these materials are removed in % °C and the maximum decomposition temperatures of each material (Figure 8b). Generally, all materials exhibiting the first decomposition step at a temperature below 200 °C correspond to the non-dissociative, physically adsorbed water molecules, moisture vaporization, as well as hydrogen bonding on the water’s surface [35]. The thermal decomposition of TP samples presents significant differences regarding the original TP. After treatment with ZnCl_2_, the TPZ decomposes mainly in one step, like TP, but at a lower temperature (Figure 8a). This lower thermal stability is associated with the incorporation of new oxygen chemical functionalities in particular carboxylic acid groups. Dehydration processes of adjacent carboxylic acid groups and/or the decomposition of these groups into CO_2_ occur at temperatures below 300 °C [51]. After combining Na_2_HPO_4_ with ZnCl_2_, it is clear that the decomposition of TPNZ takes place in two steps at around 200 to 800 °C. The first decomposition may also be due to the dehydration and decomposition of derivative groups of carboxylic acids and anhydrides, but the decomposition temperature range (from around 200 to 500 °C) is narrower in TPNZ than in TP or TPZ samples, indicating a more heterogeneous distribution of surface groups. The second carbonization step is associated with the reduction of phosphate groups by the organic cellulose, which increases the cellulose gasification in this temperature range. The char residue amount of TP, TPZ, and TPNZ were 6.96, 11.04, and 25.79%, respectively, which meant TPNZ is the highest thermal stability attributable for the phosphate groups. The maximum decomposition temperature of TP, TPZ, and TPNZ was 344, 361, and 299 °C, respectively. The T_max_ is related to the degradation of glucose rings. The T_max_ of TPNZ is lower than that of TP and TPZ, which means that the thermal stability of TPNZ was inferior to that of TP and TPZ as a result of the introduction of phosphate groups, lower crystallinity, higher cellulose content, and lower lignin content [6]. The thermal stability of TP was related to crystallinity. The more ordered molecular rearrangement led to higher crystallinity and higher decomposition temperature. Accordingly, the crystallinity of TPNZ was lower, but TPNZ is thermodynamically stable when compared with TP or TPZ. TPNZ had superior thermal stability, which also provided an advantage for its applications in heat-resistant composite materials.

## 3. Discussions

### 3.1. Action Mechanism of Na_2_HPO_4_ and ZnCl_2_ Pretreatment Reactions

As observed in SEM (Figure 4), after pretreatment with Z or NZ, the physical surface of TP and CS has been changed. The shattering of the crystalline cellulose–hemicellulose–lignin structure and the partial removal of lignin were thought to increase the surface of contact, which improved enzyme accessibility to cellulose and hence boosted enzymatic digestibility. The action mechanism of transition metal cations (Zn^2+^) in the pretreatment of CS and TP elucidate that Zn ions have been dissolved as Lewis acid, resulting in stronger interaction between the good electron acceptors, Zn^2+^, and the electron donor sites of oxygen parts of TP or CS and their derivatives, without loss of protons from hydroxyl groups of the ligand, leading to the significant weakening or rupture of C-O-C, C-O, and CH_2_OH bonds in the polymeric components of TP and CS, facilitating their breakage, consequently reducing the recalcitrance of CS and TP by fragmentation of lignin [53]. In detail, in the initial step, the coordination of the glycosidic oxygen of cellulose with zinc, which act like a Lewis acid, possibly facilitates the breakdown of the glycosidic linkage. Then, in the subsequent step, the coordinated water molecules from the hydrated ZnCl_2_ participate as nucleophiles, yielding D-glucose [34]. Kim et al., reported that the ability of metal salts to accept electrons results in varying stabilities of carbohydrate complexes [54]. The action of Lewis acids is greater than HCl and H_2_SO_4_ under the same conditions and same pH [54].

In order to compare the effect of ZnCl_2_ and its combination with Na_2_HPO_4_ on the pretreatment of CS and TP, the respective experiments were carried out under the same conditions. NZ leads to a better effect of total glucose content than that of Z (Figure 3). The overall results revealed that NZ had an especially strong effect on the enzymatic digestibility of TP and CS. On the other hand, alkaline metal cations such as Na^+^ are weak acceptors of electrons that might break the structure of hemicellulose that is not obtained by weak salts. In addition, using Na_2_HPO_4_ as the inorganic phosphorous source, the phosphorous source played a vital role in the degradation of crystalline structure (Figure 7). Moreover, the hydrolysis of β-glycosidic bonds could occur by the action of phosphate groups that boosted enzymatic hydrolysis and subsequently bioethanol production.

## 4. Materials and Methods

### 4.1. Sampling Materials

Corn stover (CS) were collected from the local farms in Beheira Governorate, Egypt, while Tetra Pak (TP) packages were collected from milk and juice sources, obtained from local companies in Egypt. Then, CS was chopped into small pieces and TP was cut into approximately 10 mm × 10 mm pieces, washed with distilled water to remove dust, and dried at 60 °C for two days until a constant weight was obtained.

The dried samples were then milled in a ball mill (FRITSCH ball milling machine, Pulveristte) at 400 rpm speed for 20 min with a 95:1 ball to powder mass ratio. The produced powder was screened using (Retsch AS200 Basic, Germany) and the particles size < 250 μm was selected. The screened CS and TP were kept in tightly closed plastic bags at room temperature till usage.

### 4.2. Pretreatment Process

#### 4.2.1. Pretreatment Using ZnCl_2_

Ten grams of CS and TP each were suspended in ZnCl_2_ (Sigma Aldrich, 99.999%, Taufkirchen, Germany) solution with a 3% (*w*/*v*) solid to liquid concentration ratio. These mixtures were autoclaved under the pressure of 310 kPa at 120 °C for 60 min using a Tuttnaur autoclave-steam sterilizer (3850 EL, Horsham, UK). After pretreatment, the mixtures were filtered, washed with hot distilled water till neutralization, and dried at 50 °C for 12 h till a constant weight was obtained. These neutralized powders were kept in tightly sealed plastic bags for characterization, enzymatic hydrolysis, and further processing, and labeled as CSZ and TPZ for corn stover and Tetra Pak, respectively.

#### 4.2.2. Pretreatment Using Modified ZnCl_2_

Both CS and TP were pretreated using a modification of the ZnCl_2_ method, through a two-stage process. The first stage is the same as that previously mentioned in Section 4.2.1, but by using Na_2_HPO_4_ instead of ZnCl_2_. Then, the produced pretreated dried powder was subjected to pretreatment using ZnCl_2_ (as procedures in Section 4.2.1). The neutralized powder which is produced after performing the two pretreatment stages was kept in tightly sealed plastic bags for characterization, enzymatic hydrolysis, and further processing, and labeled as TPNZ and CSNZ.

All the pretreated samples were characterized using physico-chemical methods such as determination of chemical composition, FTIR, XRD, SEM, and TGA (as mentioned in Appendix A), and their surface functional groups were determined (as mentioned in Appendix A).

### 4.3. Enzyme Hydrolysis

The optimum enzyme dosage for hydrolysis was determined using 0.5 gm of substrate added to acetate buffer pH 5 (1:25 gm/mL) with an enzyme dosage of 0.4 gm/gm substrate. The enzymatic hydrolysis was performed in 50 mL flasks covered by aluminum foil, incubated at 50 °C and 100 rpm for four days, the period in which a stable concentration of glucose was reached. The liberated glucose was estimated every day using a glucose kit (Bio-Med, Boston, MA, USA). The optimum enzyme dosage was subjected to CS, TP, and their pretreated samples, CSZ, TPZ, CSNZ, and TPNZ to compare the glucose yield produced after and before the pretreatment, to estimate the enzymatic hydrolysis enhancement before the pretreatment process. The released glucose yield was estimated spectrophotometrically (as mentioned in Appendix A).

### 4.4. Fermentation Process

#### 4.4.1. Yeast Screening

After yeast isolation (mentioned in Appendix A), screening of the yeast isolates was performed by using the specific yeast extract peptone agar (YEPA) medium containing chloramphenicol 0.1 g/L. Each sample was streaked on the medium and incubated at 30 °C for 48 h. The grown yeast isolates were assayed for their ethanol production efficiency by using a minimal medium of 1 gm KH_2_PO_4_, 5 gm (NH_4_)_2_SO_4_, 0.5 gm MgSO_4_, and 1 gm yeast extract containing 4% glucose by utilizing 5% yeast inoculum. Then the cultures were anaerobically fermented in tightly sealed flasks; the residual glucose was determined by using the Miller DNS method [55]. Additionally, the selected yeast was used based on the ethanol production efficiency after the fermentation.

#### 4.4.2. Yeast Cultivation

Prior to the fermentation process, the selected yeast was cultivated to proliferate the cells. Selected yeast was inoculated into a medium containing 20 gm/L glucose, 20 gm/L peptone, and 10 gm/L yeast extract, and cultivated at 30 °C and 150 rpm for 36 h. After that, the cells were transferred to the same fresh medium for further proliferation. After culturing two rounds of yeast, the cells were separated by using a centrifuge and then washed with sterile distilled water to remove any residual sugars. Finally, 50 mL sterilized water was added to re-suspend the yeast cells, and the optical density (OD) value of the yeast suspension was measured at 600 nm with a spectrophotometer.

## 5. Conclusions

This current research investigates renewable energy production for solving the problems of energy shortage that meet the United Nations Sustainable Development Goals (UNSDGs). For the first time, to the best knowledge of the authors, this effort offers a new method to produce the functionalized cellulosic materials by pretreatment of Corn Stover (CS), as agricultural waste, and Tetra Pak (TP), as an agro-industrial sustainable waste. The novel sequential pretreatment using ZnCl_2_ (Z) and combined Na_2_HPO_4_ and ZnCl_2_ (NZ) as catalysts were evaluated for improving enzymatic saccharification and consequently boosting the bioethanol production. Compositional analysis, FT-IR, SEM, XRD, and TGA revealed major structural changes after pretreatment. A combined pretreatment was systematically explored with respect to the impact of NZ pretreatment on glucose yield. The functionalization of cellulose with phosphorus and/or oxygen-containing surface groups leads to the boosting of sequential pretreatment for enhancing enzymatic saccharification of TPZ, TPNZ, CSZ, and CSNZ, and subsequently improving bioethanol production. It was also shown that the thermal stability of the pretreated TPNZ was largely improved compared to that of unmodified TP or pretreated TPZ. The outstanding enzymatic hydrolysis can be attributed to the heterogeneous surface chemistry composed mainly of oxygenated and phosphate functional groups. These findings are of great importance and clearly indicate that pretreated TPNZ represents the excellent adaptability for the pretreatment of lignocellulosic materials.

## Figures and Tables

**Figure 1 ijms-23-01777-f001:**
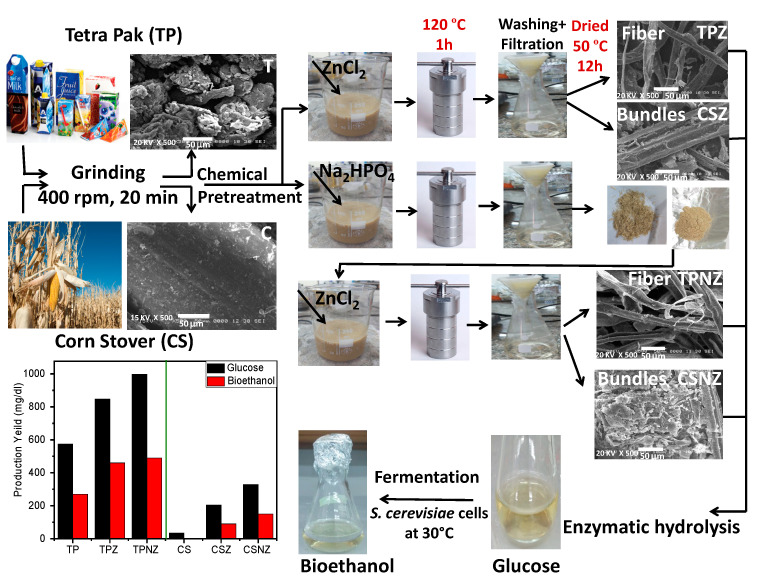
Schematic description of the pretreatment steps for Tetra Pak (TP) and Corn Stover (CS) to produce Tetra Pak and corn stover treated by ZnCl_2_ (TPZ) and (CSZ), and Tetra Pak and Corn Stover treated by Na_2_HPO_4_ then ZnCl_2_ (TPNZ) and (CSNZ) respectively, for enzymatic hydrolysis and subsequently for bioethanol production.

**Figure 2 ijms-23-01777-f002:**
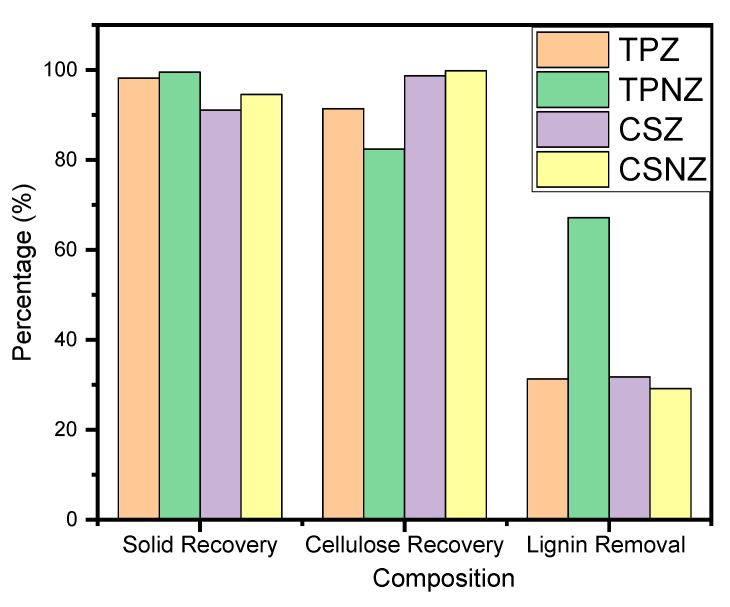
Solid recovery, cellulose recovery, and lignin removal of the two substrates under the different pretreatments.

**Figure 3 ijms-23-01777-f003:**
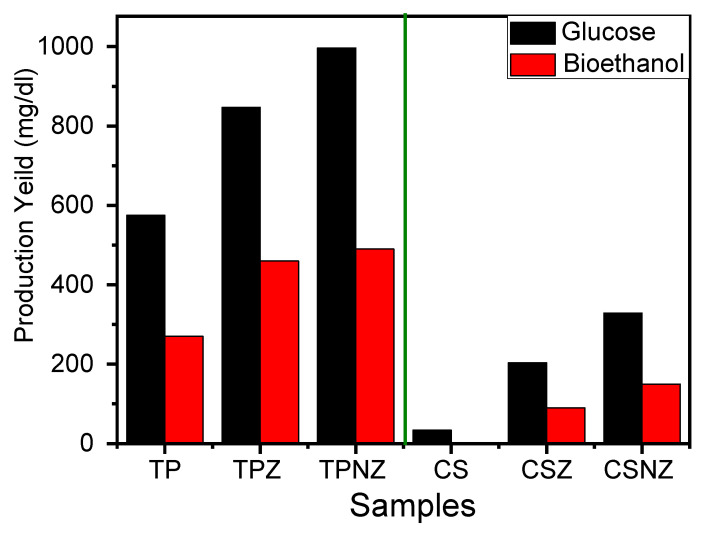
Produced glucose from enzymatic hydrolysis and bioethanol liberated after fermentation of the hydrolysate.

**Figure 4 ijms-23-01777-f004:**
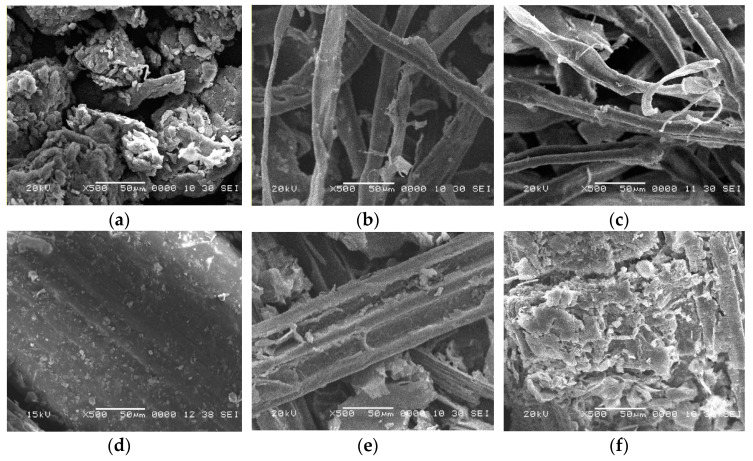
SEM images of (**a**) TP, (**b**) TPZ, (**c**) TPNZ, (**d**) CS, (**e**) CSZ, and (**f**) CSNZ.

**Figure 5 ijms-23-01777-f005:**
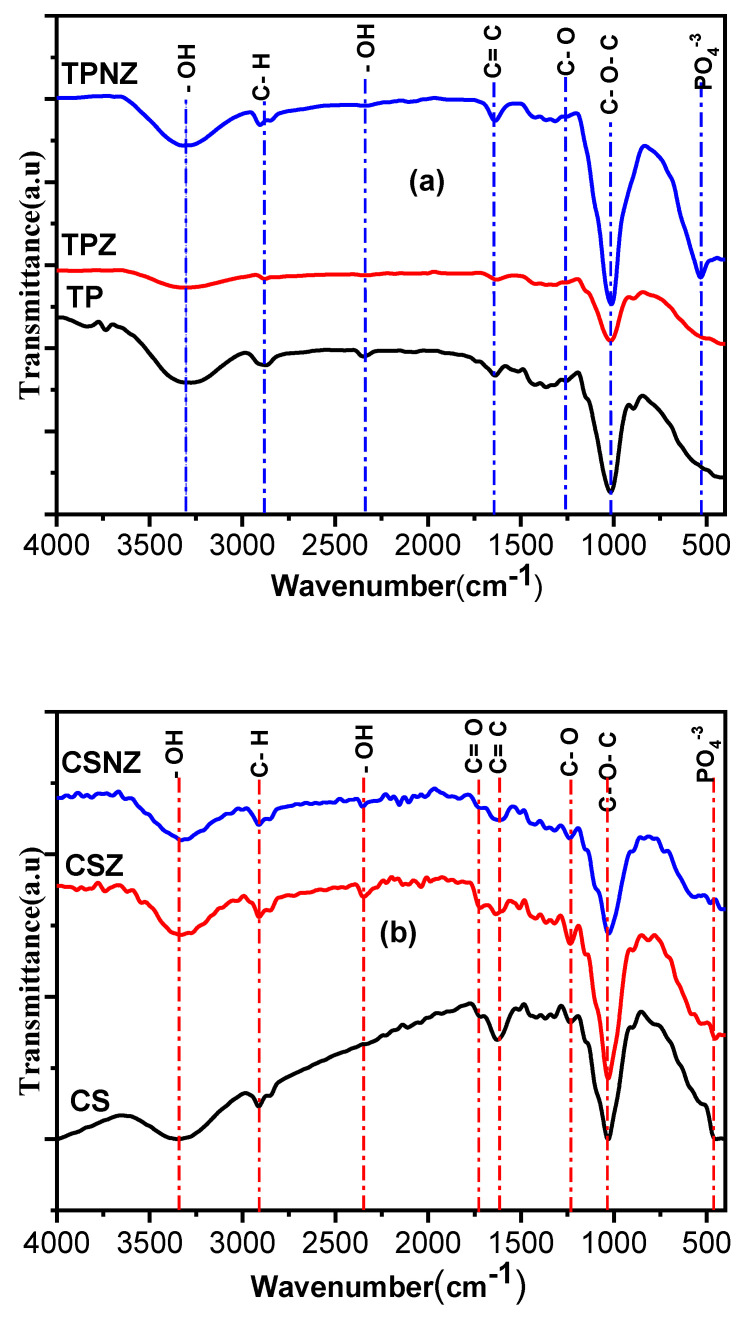
FTIR spectra of (**a**) TP and (**b**) CS, before and after pretreatment.

**Figure 6 ijms-23-01777-f006:**
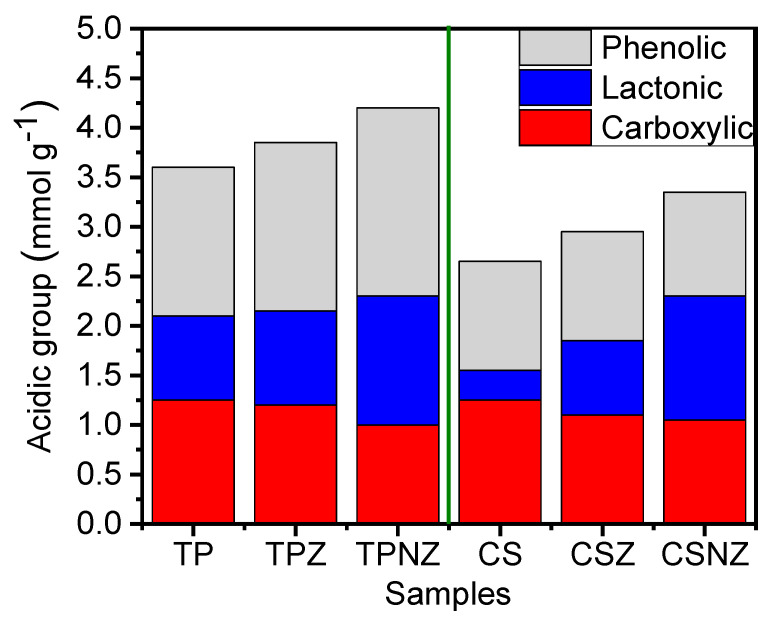
Acidic groups of TP and CS samples, before and after pretreatment.

**Figure 7 ijms-23-01777-f007:**
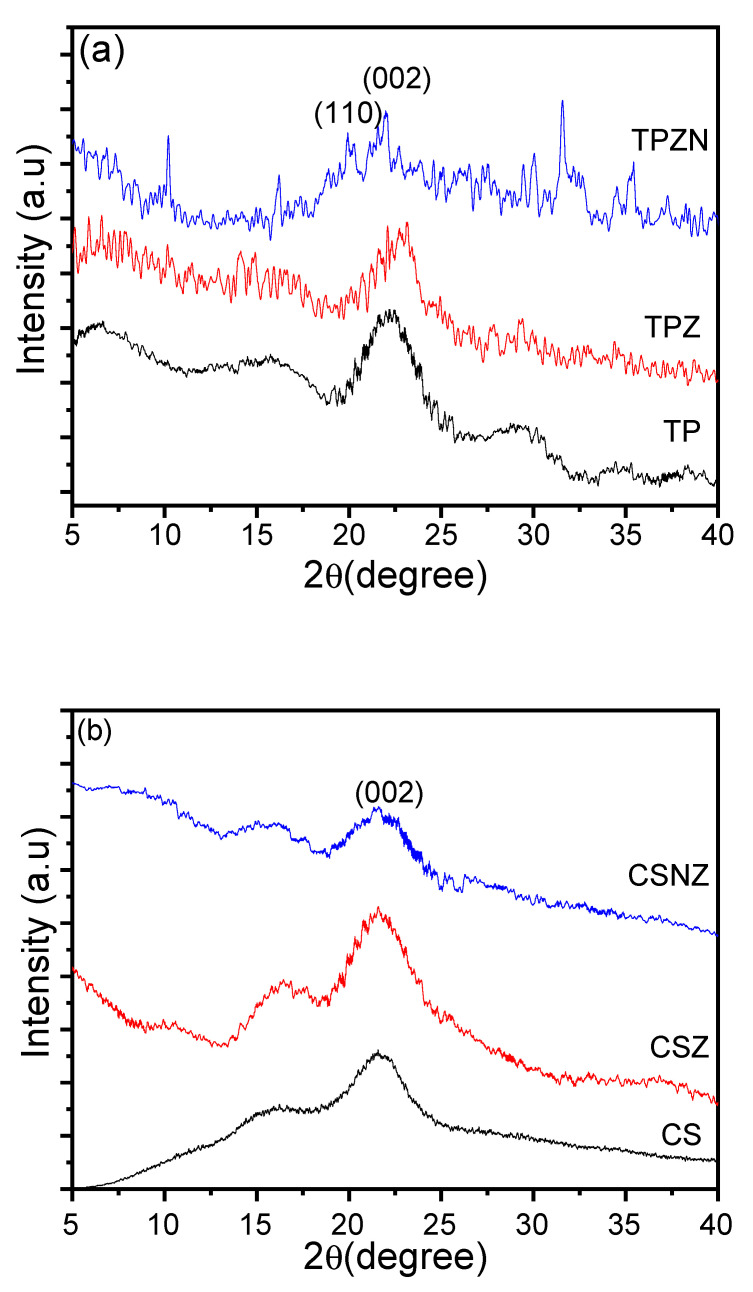
XRD of (**a**) TP and (**b**) CS, before and after pretreatment.

**Figure 8 ijms-23-01777-f008:**
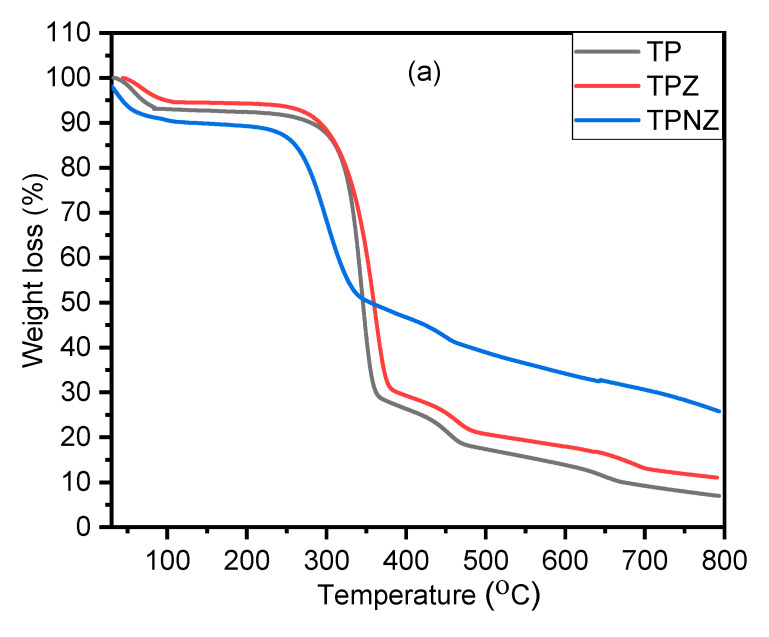
(**a**) TGA and (**b**) DTG of TP, TPZ, and TPNZ.

**Table 1 ijms-23-01777-t001:** Composition of TP, CS, and their pretreated samples.

Substrate	Analyzed Sample	Cellulose Content, %	Lignin Content, %
Tetra Pak (TP)	TP	64	10
	TPZ	59	7
	TPNZ	59	3.3
Corn Stover (CS)	CS	36	12
	CSZ	39	9
	CSNZ	38	8

**Table 2 ijms-23-01777-t002:** Comparison of ethanol production of various feedstocks pretreated with various pretreatment conditions.

Feedstock	Pretreatment Conditions	Fermentation Conditions	Biorthanol Production (g/L)	References
Tetra Pak (TP)	ZnCl_2_ and Na_2_HPO_4_ (3% *w*/*v*), 120 °C, 60 min.	30 °C/48 h	5	Our study
Corn stover	Oxalic acid (2% *w*/*v*), 120 °C, 60 min.	50 °C/72 h	3.6	[11]
Sugarcane bagasse	1.73 M ZnCl_2_, 1.36 M NaOH, 9.69% Sla, 121 °C for 60 min	30 °C/15 h	4.88	[29]
Rice straw	[Emim][Oac]-DMSO	-	1.68	[30]
Eucalyptus	NaOH (2–6%), 50–90 °C	37 °C/72 h	3.4	[31]
Wheat straw	H_2_SO_4_ (2%), 180 °C; 10 min	30 °C/72 h	0.44	[32]
Maize	Microwave-assisted H_2_SO_4_, 50 °C; 20 min	50 °C/24 h	0.51	[33]

## Data Availability

The data presented in this study are available on request from the corresponding authors.

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
