# Peer review of "Role of Combined Na2HPO4 and ZnCl2 in the Unprecedented Catalysis of the Sequential Pretreatment of Sustainable Agricultural and Agro-Industrial Wastes in Boosting Bioethanol Production"

_ijms, 2022, doi:10.3390/ijms23031777_

Round 1

Reviewer 1 Report

The manuscript proposes a new method for the production of functionalized cellulosic materials by pretreating corn straw as agricultural and tetra-pack as sustainable agricultural waste. A new sequential pretreatment using ZnCl2 and combined Na2HPO4 and ZnCl2 as catalysts was evaluated to improve enzymatic saccharification and therefore increase bioethanol production. In general, the study is detailed and informative. After careful reading and evaluation, I think the manuscript can be accepted as it stands. 

Additional comments: At the present time, despite a good deal of studies in this field, a search for alternative sources of energy is still in progress. The manuscript under review investigates promising sources such as agricultural residues and agro-industrial waste, more specifically corn straw and Tetrapak. These feedstock types are produced on a large scale and can successfully be employed to obtain high-value products, including bioethanol. The manuscript proposes new pretreatment strategies by using ZnCl2 and combined Na2HPO4 and ZnCl2. The resultant substrates and their enzymatic hydrolysis for subsequent ethanol production have been explored in detail in this study.

     The findings obtained by the authors has novelty and may spark interest among the Journal’s readership and arouse discussion on this issue.

     I do not see any critical points with respect to the study performed by the authors.

Author Response

The manuscript proposes a new method for the production of functionalized cellulosic materials by pretreating corn straw as agricultural and tetra-pack as sustainable agricultural waste. A new sequential pretreatment using ZnCl2 and combined Na2HPO4 and ZnCl2 as catalysts was evaluated to improve enzymatic saccharification and therefore increase bioethanol production. In general, the study is detailed and informative. After careful reading and evaluation, I think the manuscript can be accepted as it stands. 

Additional comments: At the present time, despite a good deal of studies in this field, a search for alternative sources of energy is still in progress. The manuscript under review investigates promising sources such as agricultural residues and agro-industrial waste, more specifically corn straw and Tetrapak. These feedstock types are produced on a large scale and can successfully be employed to obtain high-value products, including bioethanol. The manuscript proposes new pretreatment strategies by using ZnCl2 and combined Na2HPO4 and ZnCl2. The resultant substrates and their enzymatic hydrolysis for subsequent ethanol production have been explored in detail in this study.

     The findings obtained by the authors have novelty and may spark interest among the Journal’s readership and arouse discussion on this issue.

    Point 1:  I do not see any critical points with respect to the study performed by the authors.

Response 1: Thanks a lot for the faithful review.                                                

Reviewer 2 Report

The manuscript by Elyamny et al. “Role of combined Na2HPO4 and ZnCl2 in the unprecedented catalysis of the sequential pretreatment of sustainable agricultural and agro-industrial wastes in boosting bioethanol production” requires revision to address major concerns.

Comments.

  1. Abstract, the instrumental data for the characterization of pre-treated biowaste are very well-known. Please highlight the significance of the present study i.e. the quantitative data about saccharification (conversion) and ethanol production (yield and productivity) improvement as compared to controls and how this method is beneficial as compared to literature.
  2. The uses of abbreviations should be carefully cross-checked throughout in abstract and in the main text separately.
  3. Line 19-20, ~wastes management causes severe environmental pollution and health damage? Please highlight such points in the Introduction section and also add some information on waste management strategies and their limitations i.e. doi: 10.1016/j.rser.2021.111491.
  4. Line 43, source of lignocellulosic materials? What purposes.
  5. Please use conventional abbreviations in many cases in the text i.e. corn stover “CS”.
  6. Introduction, Please add information on the importance or advantages of biofuels production from biowastes and the suitability of ethanol over other biofuels and alcohols from biowastes i.e. doi:10.1016/j.biortech.2020.124550.
  7. Lines 70-75, too many citations, and many strategies? This information can be polished with a few citations and please also add information on biological methods for lignocellulosic pre-treatments.
  8. The materials and methods section can be minimized significantly as many procedures are well-known and use just citations instead of brief descriptions.
  9. Section 3.1, please add detailed information about saccharification yields, ethanol fermentation yield, productivity, and conversion. Improve the discussion as to how this approach is beneficial over other conventional ethanol production literature with quantitative data comparison? i.e. doi: 10.1002/biot.201800468.
  10. As the authors say “boosting bioethanol production” Please add a few additional data on ethanol fermentation along with literature comparison. 
  11. Sections 3.3-3.7, the discussion needs significant improvement with more quantitative comparison instead of too descriptive.
  12. Overall, please add economic analysis data for the present process.

Author Response

The manuscript by Elyamny et al. “Role of combined Na2HPO4 and ZnClin the unprecedented catalysis of the sequential pretreatment of sustainable agricultural and agro-industrial wastes in boosting bioethanol production” requires revision to address major concerns.

Point 1: Abstract, the instrumental data for the characterization of pre-treated biowaste are very well-known. Please highlight the significance of the present study i.e. the quantitative data about saccharification (conversion) and ethanol production (yield and productivity) improvement as compared to controls and how this method is beneficial as compared to literature.

Response 1: Thank you for your valuable comments. The authors added quantitative data represented in the part to the abstract and the data is promising and a competitor to other studies as the reviewer suggested 

Point 2: The uses of abbreviations should be carefully cross-checked throughout in abstract and in the main text separately.

Response 2: Thanks a lot for your valuable comments. The authors corrected the abbreviation in the abstract and in the whole text as the reviewer suggested.

Point 3: Line 19-20, ~wastes management causes severe environmental pollution and health damage? Please highlight such points in the Introduction section and also add some information on waste management strategies and their limitations i.e. doi: 10.1016/j.rser.2021.111491.

 Response 3: Thanks a lot for your valuable comments. The authors highlighted these points in the introduction section and added more information about agricultural and agro-industrial waste management and their limitations associated with the recommended reference as the reviewer suggested.

Point 4: Line 43, source of lignocellulosic materials? What purposes.

Response 4: Thanks a lot for your valuable comments. The authors mentioned the source of the lignocellulosic materials and the purpose of it as the reviewer suggested.

Point 5: Please use conventional abbreviations in many cases in the text i.e. corn stover “CS”.

Response 5:  Thank you for your valuable comments. The authors made all the abbreviations in whole the manuscript as the reviewer suggested.

Point 6: Introduction, Please add information on the importance or advantages of biofuels production from biowastes and the suitability of ethanol over other biofuels and alcohols from biowastes i.e. doi:10.1016/j.biortech.2020.124550.

 Response 6:  Thank you for your valuable comments. The authors added the next paragraph in the introduction section as the reviewer suggested

 “Lignocellulosic biomass is used for second-generation biofuel production which has several positive impacts on three fundamental pivots; 1) economic pivot as it is sustainable, increases investment in plant and equipment, leads to agricultural development, increases the number of rural manufacturing jobs, and reduces the dependency on imported fuels, 2) energy security pivot as it is renewable, available, and reduces the use of fossil fuels, and environmental pivot as it is biodegradable, improves land and water use, leads to high combustion efficiency, reduces greenhouse gases (GHG), and consequently reduces air pollution”.

 Point 7: Lines 70-75, too many citations, and many strategies? This information can be polished with a few citations and please also add information on biological methods for lignocellulosic pre-treatments.

Response 7: Thank you for your valuable comments. The authors reduce the number of citations and put it as a review and also authors put details of the biological methods as the reviewer suggested.

Point 8: The materials and methods section can be minimized significantly as many procedures are well-known and use just citations instead of brief descriptions.

Response 8: Thank you for your valuable comments. The authors sent it to the supplementary information section.

Point 9: Section 3.1, please add detailed information about saccharification yields, ethanol fermentation yield, productivity, and conversion. Improve the discussion as to how this approach is beneficial over other conventional ethanol production literature with quantitative data comparison? i.e. doi: 10.1002/biot.2018004681).

Response 9: Thank you for your valuable comments. The authors mentioned a table for comparison between our work and already published in the literature and our production is superior to literature as the reviewer suggested.  

Point 10:  As the authors say “boosting bioethanol production” Please add a few additional data on ethanol fermentation along with literature comparison. 

Response 10: Thank you for your valuable comments. The authors put the additional data on ethanol fermentation as the reviewer suggested.  

Point 11:  Sections 3.3-3.7, the discussion needs significant improvement with more quantitative comparison instead of too descriptive.

Response 11: Thank you for your valuable comments. The authors mentioned a table for comparison between our work and already published in the literature and our production is superior to literature as the reviewer suggested. 

Point 12:  Overall, please add economic analysis data for the present process.

Response 12: Thank you for your valuable comments. The authors already working at this point but the time is not efficient to finish this study.   

Round 2

Reviewer 2 Report

Accept as is